# "I'm an Academic, Now What?": Exploring Later-Career Women's Academic Identities in Australian Higher Education Using Foucauldian Discourse Analysis

Matthew James Phillips * and Peta Louise Dzidic

Discipline of Psychology, School of Population Health, Faculty of Health Sciences, Curtin University, Bentley 6102, Australia; peta.dzidic@curtin.edu.au
* Correspondence: matthew.phillips@curtin.edu.au

**Abstract:** The becoming of, and being, a later-career woman academic is marked by being positioned to play a key role in the operation of the academic institution. Tensions emerge when later-career women academics are expected to balance these expectations, while simultaneously contemplating how they choose to remain, work, and identify within academia. We qualitatively explored how Australian later-career women academics conceptualise their academic identities, and the subject positions made available through their discourse. Aged between 43 and 72 years, 17 participants were interviewed. Data was analysed using Foucauldian Discourse Analysis. Four subject positions were identified—The Insecure Woman, who experienced tensions between the academic that the system required them to be, compared to the academic that they wanted to be; The Expert Academic, viewed by other individuals as the voice of reason within academia; The Reflective Academic, who reflects on, and summarises, their academic career; and The Disengaging Academic, who begins to transition out of their academic roles and responsibilities. Overall, the identified discourses created subjectivities questioning how much one has contributed to the academic setting, what it means to have been a part of academia, as well as evaluating what it means to identify beyond it.

**Keywords:** academic identity; academia; women; later-career; academic experience; gender

## 1. Introduction

In many Western countries, an increase in life expectancy has been accompanied by the aging of the workforce, which is a trend expected to continue into the next millennium (Earl et al. 2018; Oplatka 2007). An increase in the prevalence of workers aged between 50 to 65 years has been noted, rising from 27 to 33 percent (Oplatka 2007). Workers that fall in this age bracket are conceptualised as *'later-career workers'*, chronologically considered by individuals and institutions as the period where a decline in work ability and motivations are recognised (Topa and Alcover 2019). However, *'later-career'*, as well as age, has been proposed as not purely based on chronology, rather, it can also be considered in relation to subjective age, described as how these individuals perceive their own abilities, competencies, and assessments of social, psychological, and biological age, where satisfaction with, and commitment to, the institution may increase (Topa and Alcover 2019). As such, it appears important to consider how individuals experience and identify within their workplace during the later-career period, and how they may navigate personal and professional tensions and expectations within this career stage.

In academia, the later-career stage can be typically defined as the period 15 years post Doctor of Philosophy (PhD) completion (Australian Council of Learned Academies 2012). This period has been described as a time where academics have attained the status of the *'wise'* and the *'expert'*, with increasing responsibilities being held that relate to leadership and administration roles (Berberet et al. 2005). Later-career academics are expected to be effective in their practice, playing a key role in setting the culture and tone of their

particular context, indirectly determining how the next generation of academics will exist and fare within the institution (Brown et al. 2014). In the later-career stage, academics are expected to engage in roles of authority, overseeing tasks which have both strategic and operational implications that require differing forms of governance (Altman et al. 2020). Balancing such responsibilities has been proposed to be a challenging experience for the later-career academic (August 2011; Topa and Alcover 2019).

Later-career academics have, more than most, had to respond to the professional and personal challenges presented to them, informed by increasing institutional demands, teaching quality, a focus on research outputs, internationalisation, and the evolution of the academic environment (Brown et al. 2014). To progress, later-career academics have had to engage in the continual process of occupational adaptation (Cahill et al. 2022). Facets of the academic life have had to be re-organised to improve one's opportunities in academia, where one's identities in the roles of researcher, scholar, teacher, and/or practitioner, as well as embracing the academic culture and its many demands, have been central to the process of adaptation (Pepin and Deutscher 2011). Additionally, later-career academics are viewed as *'beacons of knowledge'*, where their extensive experience and wisdom is viewed as beneficial to pass onto academics in the earlier career stages (Berberet et al. 2005). Later-career academics may be expected to take on a mentoring role at a time where the responsibilities, interests, and capacities of the academic may be shifting and adapting (Cahill et al. 2022; Pepin and Deutscher 2011).

With the life expectancy of individuals extending, this can complicate the decision-making of later-career academics, particularly in relation to their retirement (August 2011). Tensions surrounding retirement are further complicated by increasing demands being imposed on the later-career academics, with being 'sandwiched' between their children's generation (who may require financial, emotional, and parental support), and their aging parents (who may need care; Altman et al. 2020). Further, later-career academics appear to be working longer (and past their retirement age) based on the significant staffing challenges that have plagued academia (Altman et al. 2020; Cahill et al. 2021). With a lack of academics remaining until they reach the later career, those who are in this stage who are positioned within this context face tensions in how they choose to remain, and work, within academia (Cahill et al. 2021). Additionally, with the knowledge of this age cohort in relation to Academe as an institution being limited, as is academics' aspirations and plans during the later-career stage, further research is warranted (Altman et al. 2020).

Research exploring how women, specifically, navigate and identify within academia in the later-career period has been limited, due to the smaller number of women academics within senior positions (Sanders et al. 2009). The observed job pattern has been illustrated in most occupations worldwide, in that the higher the job level, the lower the percentage of women (Sanders et al. 2009). As such, understanding how women academics have progressed to the later-career stage, and in some cases, have succeeded in *'breaking through the glass ceiling'* is important, particularly in terms of how this influences their academic experiences and conceptualisation of academic identities. While it has been proposed that women academics in the earlier career stages have experienced gendered discrimination, normative divisions of home-based responsibilities, tensions between the personal and professional roles, gendered stereotyping, gendered bias, and less support than their male colleagues, it has been inferred that those who occupy a senior position and/or are a part of the later-career cohort, have found some ways to either handle or neglect the impact of such issues (Sanders et al. 2009). As such, it is important to explore how women academics have journeyed through the academic context across their careers.

Overall, research has been limited on the experiences and tensions experienced by later-career women academics, despite the growing recognition to pay attention to later-career issues in response to the changing demographic and contextual changes within our society (Cahill et al. 2022; Earl et al. 2018; Sanders et al. 2009). While much research has explored the life span of women academics, scarce literature has focused on the later-career stage (Cahill et al. 2021, 2022). Further, while the focus on being the academic, the mother,

and/or the carer is important to explore, we wish to investigate how the later-career woman academic defines herself outside of these domains, particularly as they are approaching retirement. How do the tensions within the later-career stage impact on how women academics identify within, and outside of, academia? Our research hopes to explore these experiences of the notions of self, identities, and institutional governance for later-career women in higher education, how these elements interact, as well as appreciating that knowledge can be socially constructed, can influence how different perspectives can shape institutional practices and different subjectivities. Adopting a social constructionist, critical, Foucauldian perspective, this study aimed to explore the conceptualisation of academic identities for women in later-career stage academia, and to address the overarching research questions, *(1) 'How do later-career women academics conceptualise their academic identities?'* and *(2) 'What subject positions are made available in the discourses articulated by the participant?'*.

## 2. Methods

### 2.1. Design

To explore how women conceptualise their academic identities and experiences in later-career stage academia, we employed an exploratory qualitative design, guided by Foucauldian philosophy, grounded in a social constructionist epistemological position, and critical theoretical perspective. Semi-structured interviews were conducted and then analysed via Foucauldian Discourse Analysis (FDA). Foucault's methodology allowed for us to explore the discursive constructions, discourses, and subject positions that were made available, or denied, to later-career women within their academic context, where the subjectivities and positions that individuals can adopt are established by the power relations that surround them (Rabinow 1984). The Foucauldian perspective aligns with critical theory, which is useful when engaging in the critique of oppression, power, liberation, conscientisation, and the legitimisation of social practices (Teo 2015). Dominant societal values are identified, and their influence considered on certain groups of individuals, which allows for these values, and the institutions that enforce them (e.g., academic settings) to be critiqued and challenged (Teo 2015). The social constructionist epistemology further supplements the discussed perspectives by working to both reject, and critique, the perspective of there being a universal truth, and therefore, considers the many possibilities for individuals' realities (Burr 2015). Further, the need to evaluate and critique language, discourse, power, and oppression, and how these can be located within their cultural and historical context, can be considered by our approach (Burr 2015). Using these perspectives allowed us to question and critique how these socially constituted realities impact on later-career women in academia.

### 2.2. Researcher Positionality

The research team consisted of two academics from the discipline of Psychology, sharing an interest in exploring gendered experiences through research, as well as being a part of academia, a setting that consists of contextual systemic inequities. Matthew identifies as an Anglo-Australian, Cis-gender male, within the LGBTIQA+ identity, and an early career academic in a tenured teaching and research role. Peta identifies as a White-Australian coloniser, Cis-gender female, mother and carer with an invisible chronic illness, middle-career academic in a tenured teaching–research role. We attended to this influence by engaging in a pragmatic approach, where our privilege was acknowledged, and reflexivity was engaged in, to maintain rigour and authenticity in the research process.

### 2.3. Participants

Later-career women who worked in Australian State and Territory public higher education institutions were recruited. We engaged in purposive sampling as a part of a broader research project which targeted early-, middle-, and later-career women academics, navigating the staff directories of the institutions listed on the Times Higher Education (2018). One-hundred sixty-eight women academics were invited to participate in the

research, where 90 across the three career stages acknowledged their interest (30 were in later-career). Seventeen later-career women academics (ages ranging from 43 to 72 years; $M = 56.6$, $SD = 9.9$) reflecting a multitude of academic roles (e.g., Sessional, Teaching, Research, and Teaching and Research) across 13 disciplines within 12 Australian State and Territory public higher education institutions took part in individual interviews. Table 1 outlines the demographic information collected from these participants.

**Table 1.** Later-Career Women Academics' Demographic Information.

| Later-Career Women Academics ($N = 17$) | |
| :--- | :---: |
| **Age (in Years)** | |
| Mean | 56.6 |
| *SD* | 9.9 |
| Minimum | 43 |
| Maximum | 72 |
| Range | 29 |
| **Sex** | |
| Female | 17 |
| **Current Institution** | |
| Curtin University | 4 |
| Edith Cowan University | 1 |
| Flinders University | 2 |
| Murdoch University | 1 |
| University of Adelaide | 1 |
| University of Melbourne | 1 |
| University of Southern Queensland | 1 |
| University of Sydney | 1 |
| University of Tasmania | 1 |
| University of Technology Sydney | 1 |
| University of Wollongong | 1 |
| Victoria University | 2 |
| **Current Faculty** | |
| Science | 5 |
| Technology | 2 |
| Engineering | 3 |
| Mathematics | 2 |
| Medicine | 5 |
| **Current Position** | |
| PhD/Sessional | 0 |
| Teaching | 3 |
| Research | 3 |
| Teaching and Research | 11 |
| **Identifies As** | |
| Aboriginal and/or Torres Strait Islander | 0 |
| Living with a Disability | 0 |
| Culturally and Lingually Diverse | 6 |
| Diverse in Sexual Identity | 1 |
| **Current Household Composition** | |
| Lone Person | 2 |
| With Family (i.e., Partner and Child/ren) | 10 |
| With Housemates | 0 |
| Couple/With Significant Other | 5 |
| Multiple Families | 0 |

*2.4. Materials*

The interview guide was semi-structured, with open-ended questions to probe on points of interest (Morrow 2005), build participant rapport (Ravenek and Rudman 2013)

and discover in-depth, rich discourses, and subjectivities (Foucault 1972; Ryan et al. 2007). These subjectivities could then be co-constructed by us as the research team (Foucault 1972; Ryan et al. 2007). The format of the semi-structured interview allowed for us to ask our planned questions, but also prompt, and establish the participants' social context, identified as important within the interviewing process (Ravenek and Rudman 2013). The interview guide was informed by the research objectives, the extant literature base, findings from previous phases of the research, and refined by the research team. Questions related to contextual differences between the age groups, as well as experiences and strategies that assisted the later-career women academics in navigating academia. Example questions were, *"What do you wish you had known about the academic setting before you entered it?"*, *"What do you think are some of the experiences of women in the later-career stage of academia?"*, *"What do you think are some of the challenges or difficulties for women within academia?"* and *"Can you tell me a little bit about whether you have thought about leaving academia?"*. Demographic information was collected through a survey.

*2.5. Procedure*

The research was approved by the Curtin University Human Research Ethics Committee (HRE2018-0606). A participant information sheet, consent form, and demographic questionnaire were provided to participants. Semi-structured interviews were audio-recorded, conducted online ($n = 11$) or face-to-face ($n = 6$), and lasted between 40 to 90 min. Upon completion of each interview, audio recordings were transcribed verbatim and analysed. Following analysis of all transcripts, and as a part of our quality assurance procedures, participants were invited to *'opt-in'* to engage in member checking, where we sent them a summary of the key collated messages of the research, to allow the later-career women academics to provide feedback on the fairness and accuracy of the interpretations, as well as allow for the inclusion of additional reflections to the research.

*2.6. Data Analysis*

Willig's (2008) FDA six-step process was adopted to analyse interviews, which is outlined below in Table 2. Presenting the findings as subjectivities allowed for us to address the research aims and focus, which were to explore academic identities, that pragmatically could be conceptualised as subjectivities.

**Table 2.** Conducting a Foucauldian Discourse Analysis (Willig 2008).

| Foucauldian Discourse Analysis Overview | |
|---|---|
| **Step** | **Description** |
| 1. Discursive Constructions | The discursive constructions illustrated how the later-career women academics discussed the topic of the research question. This process included both explicit and implicit references to the topic as well as describing any particular issues identified within the talk. As such, we searched the data for the different ways in which the participants constructed the discursive object of the academic identity, as well as the academic experience. |
| 2. Discourses | This step allowed for us to identify the particular discursive lens that the participants were using to capture their sentiments. Each discursive construction was considered within wider societal discourses, which served to help us understand why later-career women academics discussed their academic identities and experiences in this way. |

**Table 2.** *Cont.*

| Foucauldian Discourse Analysis Overview | |
|---|---|
| **Step** | **Description** |
| 3. Action Orientations | Here, we explored what could be gained from constructing the academic identity and experience in these ways for later-career women academics, considered within the wider discourses discovered, and as such, further questioned the function and benefit of the constructions. |
| 4. Positionings | The ways of being, or subject positions, that were made available by the discursive constructions and wider discourses for later-career women academics were explored. |
| 5. Practice | The possibilities for action that were available to later-career women academics who identified with the particular subject positions explored were considered. |
| 6. Subjectivities | The possibilities for action for later-career women academics were extended on to understand how they impacted participants' academic experiences and constructions of their academic identities. |

## 3. Findings

Four subject positions for the later-career women academics were identified in this FDA—(1) The Insecure Woman, (2) The Expert Academic, (3) The Reflective Academic, and (4) The Disengaging Academic. Here, we provide commentary on how they were made available to the participants through discursive constructions and, the role of discourse, as well as explore how they function within Australian public higher education.

### 3.1. The Insecure Woman

The Insecure Woman subject position reflects a later-career woman academic who, based on her previous experiences of prejudice, discrimination, lack of opportunity, and uncertainty of the self, is positioned to doubt her contributions, value, and academic identity. Within this positioning, there was a clear focus on how gender can manifest and influence how the later-career woman academic experiences higher education. Previous experiences of having to comply to the academic standards, learning how to be pragmatic, and engaging in prototypical female roles and responsibilities positioned the women to experience tensions between the academic that the system required them to be, compared to the academic that they wanted to be.

First, the later-career women academics acknowledged that they experienced a sense of insecurity when constructing their academic identity, specifically, of what the academic system, as well as other academics, expected of them. While the women drew on patriarchal discourse to contextualise how the conditions of academia were constructed to position them as outsiders, discourses evident of survival and risk also manifested within their experiences. Being in a position where they were feeling unsure of the expectations of academia meant that the later-career women doubted their capability to contribute to the setting: *"…there's a mismatch between what I think the expectations are, and what I think the expectations should be, and that makes me doubt myself and what I've contributed to the institution. Am I seen as a genuine, proper academic?" (04).* This sentiment supports the manifestation of survival and risk discourses in the way that the Insecure Woman identity is constructed for later-career women academics, who discursively constructed insecurity as a part of their academic identity. The primary consequence of this construction is that the women may question their own abilities and contributions, as well as their perception of what it means to be an academic. Additionally, the women appear to question whether they have contributed enough to be evaluated and considered as a *"proper" (01)* academic, where the women may find that there are inconsistencies between their own insecurities, and the institutional expectations of being an academic.

For some of the later-career women academics, feeling insecure about their academic positioning and navigating the system perpetuated feelings surrounding their struggle in what it means to be a woman in academia. Nuance surrounding gendered roles, identities, and positionings manifested when the later-career women drew on their academic experiences with patriarchal discourse to explore how women were viewed as the minority in academia, even within the later-career stage: *"I'm a woman in academia, and because of that, I'm part of the minority. Women are always the minority in academia, and in everyday society. Blame the patriarchy" (09).* Discussions with the participants illuminated how for some, patriarchal discourse appeared to embed itself within normative practices and assumed responsibilities over the course of the academic career. For these women, they experienced assumptions made of them, their abilities, and difficulties based on gender, which further perpetuated their insecurities:

*"I think it is difficult for women to progress their career, because of, you know, the structures within the university, to be able to do that, and be viewed as an academic, rather than as a woman, or a woman academic. I have found it extremely difficult to progress based on my gender. It is as if I may as well have a sign on my head that says 'impediment' just because I'm a woman." (06).*

Based on these assumptions and experienced difficulties, the later-career women academics felt as if they were evaluated based on their gender, and judged based on their physicality and biology, rather than their intellectuality. Gendered/heteronormative and patriarchal discourses continued to manifest throughout the experiences of later-career women academics, where the women felt unsupported based on both systemic disadvantages, and their gendered and caring responsibilities. Additionally, the women felt that having more responsibilities to juggle, compared to their male academic colleagues, positioned them to struggle in relation to their progression in academia:

*"I look at most of the people who do my job, most of them are male. . .most of these men are all married with children, and they have wives to help out. I'm trying to do the job that they're doing, but I'm also doing the job that their wife is doing as well. . ." (15).*

Here, the participant draws on gendered/heteronormative discourse to explore how later-career women academics navigate and juggle the increasing expectations set for them in both the professional and personal context. The participant described how her responsibilities were effectively doubled based on the gendered expectations that manifest throughout their experience. Male counterparts are described as being able to be an academic, whereas the participant describes having to be an academic, and having to complete the duties performed by her counterparts' wives. When reflecting on this nuance, consistent with the early- and middle-career stages, gender appears to manifest throughout the experience of women, particularly in how they can occupy and navigate academia, and juggle caring responsibilities, which men appear to be excused from. For some women in the later-career stage, this experience appears continual, which positions them in such a manner where tensions surrounding gender will consistently be at the forefront of their experience.

The later-career women academics suggested that academia was continually changing in terms of how one is expected to learn, engage, and adopt new ways of being. Additionally, the later-career women felt it was assumed based on their career stage, and age, that they may find it harder to adapt to change. Societal and ageist discourses were evident when the participants shared their sentiments supporting these claims:

*"It's like they look at me as this old, stale woman, never wanting to change, still with a Nokia phone, that they have to manage. It's like 'I have an iPhone, and I can learn new things, just be patient with me!'. But that's society, isn't it? Old people are viewed as resistant to change, or that they can't learn new things, and I'm lumped in that category now." (12).*

Consequently, the above sentiment may reinforce stereotypes surrounding age and change, as well as maintaining the perspective that younger academics are able to adapt more easily, and learn new ways of being, compared to later-career women academics. Here, the women may struggle with conceptualising the self and their academic identity in the later-career stage. The constructions and discourses here function to position later-career women to not feel good enough, or to feel as if they are not the best fit for academia. In practice, it appears that for later-career women academics to mitigate the feeling of insecurity, that they must be responsible for their own learning, as well as adapting to the system, or risk feeling like an imposter within their institution.

For some of the later-career women academics, feeling insecure, dispensable, unsupported, as well as struggling to adapt to the setting, meant that they began to question their role and identity as an academic. Subsequently, the later-career women expressed fears surrounding the insecurity of their existence, and their positioning, in academia, and that they would be forced to adopt behaviours that were considered as normative to the traditional academic. These behaviours were constructed to disadvantage the later-career women in academia:

> *"Sometimes, I have no idea what the institution expects of me...I've been told that I don't meet the imposed targets that they [the academic institution] want me to meet, to be considered as ideal, and a good worker...[it] makes me question my whole being as an academic, yet I'm expected to fix the problem and to stick it out to remain a part of the institution." (17).*

Here, the participant suggested feeling as if their academic identity and way of being was not good enough, or *"ideal" (02)*, for the academic system. Survival discourse was drawn upon when considering how the Insecure Woman is expected to *"stick it out" (03)*, stay strong, and be resilient in a setting where their identities are questioned and compromised. This excludes the ways of knowing, being, and doing that are women centred. Consequently, responsibility is deferred to the self in terms of how to manage insecurity and the questioning of identity within academia, where the later-career women expressed feeling as if they need to be able to survive by resolving their own insecurities. In practice, this perpetuates their insecurity as an individual problem, rather than being influenced by, and the product of, the existing institutional system, in relation to how it manages its members.

Within the Insecure Woman positioning, the later-career women acknowledged that their feelings of insecurity were about more than just constructing a way of being that met the expectations of the institution. The constant feeling of insecurity permeated through how the later-career women academics managed their multiple identities: *"By feeling insecure at work, I begin to feel insecure at home. I bring it home with me, and I begin to question everything I'm doing" (10)*. We propose that the Insecure Woman positioning is more than just feeling constantly insecure, where additionally, it becomes a dangerous embracing across the many identities and ways of being for the later-career women. By being unsure and insecure of their academic identity, we propose that there is scope for the women in this position to be vulnerable to the threat and danger surrounding women's experiences in academia. Specifically, the way of being here could make the women more predisposed to conceptualising an identity that benefits the academic institution, rather than the woman herself, to avoid risking their own position in academia. As such, the assumption here is that to survive and maintain their positioning in academia, later-career women academics who identify as insecure must sacrifice part of their own individual, self-identity, and way of being, to be viewed as *"ideal"* and good enough for the system. The tension evident for the Insecure Woman is that she wants to remain unique and construct her own way of being, rather than being a product of academia, but she remains uncertain of exactly what her way of being is.

### 3.2. The Expert Academic

The Expert Academic subject position depicts a later-career woman academic who is constructed as a part of the most qualified group of individuals within the academic institution. Within this positioning, it was evident that the career stage of the academic, as well as their extensive academic experience, positioned them to be viewed by other individuals (academic, or otherwise) as the expert, and the voice of reason within academia. What was of interest when exploring this subject position, was the change in discourse and rhetoric surrounding the labelling used. It was evident that the women, based on their career stage, were now being evaluated and positioned based more on their expertise and experience, rather than their gender. While gender was not completely removed within the subject positions, some of the participants felt a shift in how they were evaluated, and that being positioned in this way signified to them that their journey was ending, and that they were perceived as having *'made it' (14)* within academia. This implies that within the academic experience, one must persist and survive until the end to be viewed as the Expert Academic.

First, the Expert Academic subject position was illustrated through the later-career women academics discursively constructing what it means to be viewed as *"the expert" (09)* within academia. For example, the women reflected on how they were now evaluated in relation to their expertise, rather than their gender, which had been at the forefront of their previous experiences. As such, the focus on experience and expertise positioned the later-career women as *"fully fledged proper academics" (05)*, drawing on expert, and gendered/heteronormative, discourses to illustrate the shift, as well as how they were perceived as the most qualified in academia:

> *"Previously, it was all about being a woman, and how I had to juggle multiple responsibilities with my work and home life. Now, it's almost like, because I've been here for so long, and I've shown that I can do that successfully, that the ship has sailed, and I'm now viewed solely on what I offer. I've given a lot to the institution. It's like I've made it here, finally, I'm a fully-fledged proper academic, and it feels great." (05).*

Here, the participant explores how in evading being a woman within the academic space, they were then able to be constructed as an academic. The primary consequence of this construction is that participants have been evaluated differently in academia compared to the early and middle-career stages, which may imply that experience, career stage, and status impact the perception of women within academia. Through this discourse, the assumption here implies that to be an expert, one must be older, wiser, and embedded within the setting for a prolonged period.

Additionally, the later-career women academics acknowledged that being constructed as *"the expert" (09)* appeared synonymous with being the voice of reason, and that the vast array of experience positioned the academic as effective problem solvers who knew how to overcome *"any struggle in academia" (07)*. The discussion surrounding expertise, experience, and wisdom with the participants illustrated that for some, expert and age-related discourses were embedded within the day-to-day experience of the later-career women academics. For these women, assumptions were made of them based on their career stage, perpetuating stereotypical assumptions of older adults in society, where to be considered an older adult, one must be wise, they must have all the answers, and they must be experienced: *"As an older academic at the later-career, people naturally assume you're a brilliant researcher, or a brilliant lecturer, or a brilliant manager of people. It's like you're old, and you're wise, so you know everything." (13).* Consequently, this may place unintended pressure on the later-career women if they are not able to resolve issues or meet the expectations of the Expert Academic subject position: *"Not meeting the expectations of being older and wiser, well, that's a bit daunting for me." (13).*

For some of the later-career women academics, being positioned as an Expert Academic within the institution allowed for them to feel more certain and confident in their academic positioning. The women acknowledged that in navigating academia over time,

that the factors that inhibited them from doing what they wanted in their careers, were no longer an issue. As such, they felt assured in terms of how to juggle their multiple personal and professional responsibilities. This was also easier based on their children being older and able to look after themselves, having less responsibilities from the personal domain. Security and expert discourses were drawn upon to explore these sentiments:

> *"I've made it, and I feel so good in myself as to what I've done and how I view myself as an academic. I'm viewed by others based on what I've done and how I've contributed to my area. I feel like I can do whatever I want. What can they [the academic institution] do to me? Fire me? They'd never do that; how could they survive without me?" (08).*

Here, based on the discursive construction of confidence, certainty, and security evident in identifying as an Expert Academic, it appears that the primary consequence is that the later-career women academics are positioned to feel more empowered within their identity conceptualisation, as well as in their navigation of academia. The later-career women acknowledge feeling less concerned about contributing to, and making it in, academia, as they have been able to determine exactly what and how they have contributed to the setting. Being able to conceptualise the academic identity within this career stage appears to determine just how secure and confident the later-career women academics feel in terms of their academic positioning.

Further consolidating the Expert Academic identity for the later-career women academics was being acknowledged and respected as an academic in relation to their career status, expertise, and experience. The later-career women academics suggested that as they have established their credibility based on their many years of experience in academia, that they were more respected in relation to their academic expertise. Specifically, the participants suggested how the discursive construction of a *"good academic" (11)* manifests when considering expertise, experience, and respect:

> *"You're more able to withstand some pushback, and you have established your credibility in the academic world, and so, you know, people will actually pay you respect and acknowledge that you have the authority and the credibility to say and do things. I'm viewed as a good academic, and I'm respected based on my contributions and what I offer, which feels great." (11).*

Here, the participant acknowledges that being positioned as an Expert Academic offers the benefits of credibility, respect, and authority. Based on the above discursive constructions of expertise, confidence, experience, and respect, the later-career women academics also acknowledged feeling as if their academic position was less threatened, and that they were protected from some of the negative consequences and outcomes that existed within academia based on their status. For example, a participant drew on a survival discourse to explore the trajectory of the women's academic experience, and how the Expert Academic positioning affords protection against some of the disadvantages of working in academia:

> *"I hear the stories of my colleagues, the women who are just starting out, and the women who are right in the thick of it, you know, 'the struggle', and I feel awful for them. I've experienced it. I've been there, but I've come out of it, I've survived. It's almost as if because I've survived it, I'm protected. Nothing can penetrate me, nothing can break me down, I'm safe here [in academia] now." (01).*

Here, the survival discourse manifests within the participant's sentiments, where it appears that the navigation of the academic journey as a trajectory is demonstrated within the Expert Academic subject position. Of particular interest here, is the construction of *"survival" (02)* where the participants suggest *"nothing can break" (07)* them, acknowledging the violent nature of academia for women, and implying that in experiencing the worst throughout their academic career, they have navigated an array of differing experiences intended to disadvantage and *"break" (07)* them, as well as being considered an expert in navigating these experiences. We propose that this perpetuates the problematic messaging

that later-career women academics are forced to navigate the complexities of the academic setting which are based on discriminatory practices, threat, violence, and sacrifice, and that this navigation is justifiable based on the experience of surviving said setting.

Within the Expert Academic positioning, the later-career women academics also acknowledged that the increasing amount of flexibility and autonomy afforded to them meant that they were able to spend more time reflecting on their progression, as well as considering what their next steps would be both in academia, and beyond their career. Discourses surrounding preparedness and agency were evident where the participants contended with tensions relating to their desire to remain in academia, versus planning for the future and beginning to consider whether they still want to be a part of their institution:

> *"I'm really starting to question what my future looks like in academia, and even after academia. I mean, I've achieved everything that I possibly wanted to achieve, and even more than I could have ever imagined, but it's like now what? If I stay in academia, I worry I'm going to be stuck, stagnant, not knowing what I'm doing next, and doing the same thing day in and day out. I have thought of leaving, I feel ready to leave, and I know I have the ability to make this decision. It's just hard when it's all I've ever known. How do I even begin to prepare myself for this?" (16).*

Here, the participant acknowledged how they were able to achieve in academia over time. Subsequently, reflecting on their journey illustrated tensions surrounding what it means to be an academic when moving beyond the academic system. When reflecting on their academic positioning and identity, the women explored a direct consequence to being considered as the Expert Academic. Being positioned as the Expert Academic here assumes finality in relation to the academic journey, and that at a later point in time the later-career women will feel ready to exit the system. As the participant explores above, this process is not necessarily as straight-forward as expected, when considering what it means to be an academic beyond one's career. As such, the tension evident for the Expert Academic is that she wants to remain a part of academia in her own way, but she remains unsure of what else she can contribute to the setting.

### 3.3. The Reflective Academic

The Reflective Academic subject position depicts a later-career woman academic who begins to reflect on, and summarise, their academic career, and considers whether they are satisfied with what they have achieved over time. Within this positioning, it was evident that the women were experiencing tensions surrounding their decision to continue working in academia. This was based on their own reflections surrounding their impact, versus considering how other academics perceive them in relation to their contribution to academia. As such, the later-career women academics reflected on their own positioning within academia.

First, the Reflective Academic subject position was illustrated through the later-career women academics discursively constructing how they have made an impact within academia. To engage in this, the women drew on stories that summarised their academic experience, as well as acknowledging whether they felt satisfied with the impact they have made, based on their academic contributions. As such, reflecting on their impact in academia positioned the women to quantify, or measure, exactly how they have contributed, drawing on neoliberal/economic, and responsibilisation, discourses to illustrate this process:

> *"I would think that I'd be known as a person who had an impact on others, in terms of personal impact, but also, an impact in how we think, how we learn. I've conducted some pretty novel, and cutting-edge research, and I've got an endless list of research outputs, which in the setting, shows that you've done your thing, and made a solid impact in your area. Also, knowing that my efforts are valued and appreciated is the best. Knowing that I have made an impact on even one student, or one colleague, and how they see the world,*

*and how they think, is enormously rewarding. That's what we want to see as academics, that's partly why we are here, doing what we do." (14).*

The above sentiment supports how later-career women academics are discursively constructed based on their perceived personal, and professional, impact. Neoliberal/economic and responsibilisation discourses were embedded within these constructions to explore how the women are positioned to operate within the academic setting. While the participant reflected on their desire to make an impact within the setting, the primary consequence of this construction is that they are primed to quantify their impact within academia, measured through their success and research outputs, as well as changing the perceptions and worldviews of their students and colleagues. It is assumed here that academics are expected to have an impact within the setting, and that it is part of their academic responsibility to illustrate how this has been achieved. As such, we propose that this could embed a perceived pressure within academia which enforces a need to make an academic impact. This pressure could be further heightened when considering how the expectations of neoliberal academia can impact on women's experiences and identities, where women may be under a considerable amount of pressure, compared to their male colleagues, to demonstrate their impact.

Additionally, the participants acknowledged that a part of the later-career stage was engaging in constant, day-to-day reflection on how they have made an impact in academia. The women extended on this and reflected on how they were positioned to give their perspective on professional matters, but additionally, how they wanted to assist others and provide their knowledge and expertise to be able to make a difference within the setting. Expert, as well as age-related, discourses were drawn upon to explore these sentiments:

*"As I get older, I get approached by more and more people to give my perspective on academic matters, as well as assisting some of our younger academics in their teaching and research endeavours. I quite like doing that because it is fulfilling for me. Part of it I feel is based on my expertise, and part of it I think is based on the stereotypical 'I'm old so I'm wise' perspective, but hey, if it's helpful to them..." (04).*

Here, while assumptions were made of these women based on their career stage and age, the participant considered being able to pass on their knowledge, as well as assisting other academics and being viewed as making a difference in academia as an important part of their academic experience. Consequently, the participants are afforded the agency to assist and mentor others, as well as reflect on their academic experience in a positive manner. Additionally, by being positioned to reflect positively on the impact of their academic experience, the later-career women academics acknowledged how they felt as if they had come *"full-circle" (06)*, and that they began to consider what the future held for them within academia:

*"I've had an amazing career. I've been able to meet all the expectations. I've been able to assist others and collaborate with some wonderful people. I feel like I've done all I can do here in academia, I've come full-circle, and now I'm starting to think about what else I can do here in the setting." (06).*

For some of the later-career women academics, being positioned as The Reflective Academic, and acknowledging that they had come *"full-circle" (06)* in their career prompted thoughts surrounding their future in academia. Specifically, the women acknowledged that they had begun to reflect on the suitability of their academic role for them in the future, navigating tensions between feeling passionate about their work, versus experiencing boredom based on the repetitive nature of their roles and responsibilities. A participant expressed feeling bored in their later-career stage and questioned whether they should stay within academia based on passion, versus their desire to leave the setting based on having *"made their mark" (10)* in the institution. Survival and preparedness discourses were drawn upon to explore this tension:

> *"People know I've made an impact here [in academia]. I've made my mark, and I've done all that I can do. I'm not sure what else I can do right now, and it's getting a bit boring and repetitive for me. I feel like someone else can take the reins and keep on going, but that person is not me. I've done my bit, and I've been thinking that it's time for me to go and leave academia."* (10).

Here, the participant reflects on their academic experience and feels positioned to take a step back based on their perceived impact in the setting over time. For the women, their reflective process has resulted in acknowledgement of the finality of their academic career, and that embedded in a discourse of preparedness is the consequence of feeling ready, and preparing oneself, to leave academia. The later-career women academics acknowledged feeling less concerned about contributing to academia in the future, as they express knowing exactly what they have done, and how they have made their impact in the setting. Being able to conceptualise their positioning appears to determine just how secure and confident the later-career women academics feel in terms of their future beyond academia.

Upon reflecting on their academic impact, the later-career women academics suggested they were perceived by others as being selfish if they chose to remain in their academic role, acknowledging forms of ageism in their experience, in relation to how other academics viewed them as needing to exit academia, based on their older status and access to opportunity over time. It was perceived that the academic role would be better suited and positioned for a younger academic. The participants suggested how the discursive construction of a *"selfish and greedy"* academic manifested when reflecting on whether the later-career woman academic decides to remain in academia:

> If there was one thing I would complain about in my old age, it is the subtle and also overt reference of my age and its relation to my position here. I've had colleagues ask me, "So why haven't you left yet? You've done all you can here for a very long time. Give someone else a go.". Someone else said to me more explicitly that I'm "selfish and greedy for remaining here when it's time for me to go". I can understand where these perspectives are coming from, as academia is extremely competitive, but like, I gave it my all to get here. Don't start telling me when I should be leaving." (08).

Here, the participant explores how one's career stage, age, and experience can impact other academics' perceptions of how long one is meant to remain in academia. We question the imposed deadline being placed onto the participants and propose that the pressure placed onto later-career women academics to leave academia when they have reached a certain time in their career places negative connotations of blame and shame on them for occupying a position that they have worked hard to achieve. Further, ageist, risk, and neoliberal/economic discourses were evident when the participants acknowledged how some other academics perceived their positioning in the later-career stage, expressing that while they felt it was their time to leave academia, they could not help but feel this was influenced by broader perspectives relevant to age, which in turn, influenced their own perspectives:

> *"I want to be able to leave academia when I feel the time is right. I can tell you that I feel like my time to leave is now, or at least, sometime in the next few months, but I question whether I came to this decision on my own. We are told day in and day out that the institution wants hard workers, who meet the forever changing expectations of academia, who are able to work day in and day out, yet, as an older academic, I feel that other academics, and the system also, assume that we will be slowing down, because we are old, and that our output will shift and change, which then, for us, is risky, because the institution then perceives us as not being able to bring in what is expected in our role. With this perceived shift, comes the subtle reminder that we are old, and it's time to go."* (17).

It is evident that broader societal views of what it means to be an academic, as well as what it means to be an older individual, construct tensions within the later-career academic experience. Shifts in what the academic system expects of an academic were expressed

in the above sentiments, embedded within a neoliberal/economic discourse relating to academic outputs and expectations of working in academia, which conflicts with the expectations and perceptions of being a later-career woman academic. In practice, this impacts on the ability for the women to have agency and autonomy in deciding when to leave academia. Later-career women academics can be placed in a precarious position here, where the choice of leaving academia paradoxically denotes a lack of choice, with the above choices appearing conditional. It appears that the later-career women academics experience these tensions when reflecting on their academic experience, compounded by the unspoken given that they would relinquish their academic position when reaching a certain age or status within the setting, based on broader societal views of competency and age.

Based on the above tensions, the later-career women academics acknowledged they felt as if they were "*taking up space*" *(03)* within academia, which further contributed to their consideration of what their next steps would be. As such, when reflecting on their academic experience, considering their personal and professional impact, their suitability and passion for their work, as well as other academics' perceptions of their academic positioning, the later-career women academics acknowledged a shift in the way they approached their roles and responsibilities. Discourses surrounding preparedness, agency, autonomy, and survival were evident, where the participants expressed their desire to leave academia, and that no matter what, or who, was influencing them, it was their time to go:

> *"I've begun thinking about how I'll be feeling on the day I leave. I've spoken to my line manager and the head of school and expressed that I feel like it's time for me to go. I want to leave on my own terms, and I honestly feel that I've made my own decision. Next January, I'll sign off on an amazing 29 years in academia. I've survived, and now it is time for me to do a little less surviving, and a lot more thriving." (12).*

Here, the later-career woman academic reflected on their perceived agency and autonomy in deciding to leave academia. Subsequently, the reflection here illustrated similar nuances to the Expert Academic subject position. For example, when reflecting on their academic positioning, and identity, the women explored a similar direct consequence and assumption of finality to one's academic journey. As the later-career woman academic feels ready to leave academia, the experience is left open-ended, where the participant begins to acknowledge what it means to be her own individual beyond the academic setting.

*3.4. The Disengaging Academic*

The Disengaging Academic subject position is a later-career woman academic who, while feeling rewarded in what they have achieved in academia, as well as the impact that they have made, begins to transition out of their academic roles and responsibilities. Within this positioning, it was evident that the later-career women were experiencing tensions surrounding what it means to leave academia. This was based on feeling ready to leave the setting but acknowledging that it would be difficult to remove themselves from a role they have been embedded within for so long. As such, the later-career women academics reflected on how it would be a complex process when identifying and positioning themselves outside of academia, as well as having to consider what it means to be an academic who will have left the institution.

First, the Disengaging Academic subject position was illustrated through the later-career women academics discursively constructing their feelings surrounding who they are as an academic, and what they have achieved over time. The women shared with us stories that summarised their academic experience, expressing that they felt satisfied with their academic role, as well as who they were as an academic within this career stage. As such, feeling satisfied positioned the women to be able to move away from, rather than closer to, the demands of the academic institution. Survival discourse was drawn upon to illustrate this:

> *"I feel so good about what I've achieved here at [name of institution]. I've done it all and because of that, I'm slowing down, and bowing out. I don't care as much about the demands of academia anymore because I've done everything I wanted to do, and more. I'm not here to survive, to be a part of the setting, and to try and fit in anymore." (15).*

Another participant expressed similar sentiments when exploring how they have begun to disengage from academia, based on their feelings of accomplishment, achievement, and success. Specifically, the participant considered how the disengagement process acts as a manner of preparing oneself for the difficult separation of self from academia, drawing on preparedness discourse to illustrate this experience:

> *"I'm doing less work every single day, and that's deliberate. If I take on less and less, then I feel as if I'm preparing myself for the eventual time that I'll have nothing to do. Each day of most of my adult life has encompassed some kind of academic task or responsibility. Not having that kind of responsibility every single day is scary to me, but I feel like if I try and remove myself from it a little bit more each day, then I think I'll be more prepared for when I eventually leave here [academia]." (16).*

The above sentiments support how the later-career women academics discursively construct their success and achievements in relation to their academic identity, as well as how they construct the later-career stage as a time where an academic begins to disengage from their roles and responsibilities. Both survival and preparedness discourses were embedded within these constructions to explore how the women were positioned to operate as they came to the end of their academic experience. While the participants acknowledged acceptance of their time in academia, feeling satisfied, achieved, and rewarded based on their academic contributions over time, it was evident that they had begun to disengage from their responsibilities as a way of preparing to leave academia. The consequence of this disengagement is that the women are not only lessening how responsible they feel in relation to their roles, but additionally, these actions allow for them to prepare for when they leave academia. As such, we propose that disengaging from academia in this manner could be a useful and helpful mechanism for the later-career women academics to slowly shift and change how they are positioned in the setting, limiting the potential negative impact that could result. This could also be helpful for all individuals who are considering retirement.

With the later-career women academics acknowledging their readiness to transition out of academia, they expressed how experiencing boredom and monotony surrounding their academic roles only further emphasised the desire to leave academia: *"Oh Matty, I love it here, but I'm getting so bored! Thank goodness I'm leaving soon! Retirement here we come!" (01).* With experiencing this boredom, and moving away from the demands of higher education, the later-career women academics expressed how they felt the most able to shape their academic identity at this time, compared to the other career stages. One participant spoke explicitly in terms of how their lack of desire to progress further (based on the perception of already *"making it to the top" (04)*), as well as no longer valuing the mundane and routine nature of academia, allowed for an increased amount of autonomy and agency surrounding how they engage in their role. For the participant, this validated their capacity to disengage, drawing on expert, as well as agentic, discourses to explore this nuance:

> *"Let me tell you, I'm finished with the tasks that bore me. The repetitive shit that we have to do, day in and day out. What I've noticed is that compared to 25 years ago when I first started in academia, or even 10 years ago when I was fairly seasoned but still had a long way to go, is that nowadays, I can have a lot more say in what I do. I don't have to do that shit anymore. I've made it to the top here at [institution's name], so I can decide who I am and what I do here. Also, I'm leaving next year, and they know that, so I lowkey don't really care so much anymore about it all." (04).*

The above sentiment explores how this later-career woman academic has been able to determine if, when, and how she engages in her academic responsibilities, based on her planning of, and decision to, leave academia. Expert, and agentic, discourses were evident

in terms of how the participant expressed having *"made it to the top" (04)* in academia, and consequently, being viewed as someone who has choice and control in the setting informed how she navigated her capacity to disengage. As such, the later-career women academics may be afforded more capacity to disengage based on (1) their career status and perception of expertise, and (2) their decision making surrounding their choice to leave academia. Being positioned in this manner allows the later-career women academics to shape their own academic identity, specifically as someone who is choosing to disengage from this setting.

For some of the later-career women academics, disengaging from the academic setting, and separating the personal self from the academic self, illustrated a difficult process. Specifically, the women shared with us stories of their academic experiences and acknowledged the diversity and depth of their experience over their many years being a part of academia. Tensions emerged when the later-career women shared their concerns surrounding how difficult it may be to separate themselves from a role, and an identity, that has encompassed who they have been for so long: *"I've been an academic for so long. Who am I beyond that?" (02).* Extending on these claims, the participants felt that part of the difficulty in separating from academia is that they were preparing themselves for a new experience, and a new identity, that was not yet clear to them, feeling forced to reflect on what was to come beyond academia:

> *"I've been an academic for so long, that I can't remember a time where I wasn't one. I'm ready to leave, and I'm proud of what I've achieved, but what do I do beyond here? Our lives are so 24/7 academia that knowing what to do outside of the setting is tough. We never really get the time to have outside interests, but now I feel like I need to have a sense of what's to come, when really, I haven't the faintest idea what to expect." (08).*

The above sentiments explore how the later-career women academics consider their progression beyond academia, where they acknowledged feeling the pressure to prepare and construct a new experience and identity. This process was acknowledged as complex, and while the women drew on preparedness discourse to explore how they would navigate the process, this was still considered difficult as they expressed feeling unsure of how to identify beyond academia. This was based on how both overworking within the system and experiencing the lack of time to engage in other roles and responsibilities, can complicate the process. We propose that this may force the later-career women academics to reflect on what comes beyond academia at a time where they may not feel ready to consider it. Additionally, the later-career women acknowledged that while they had disengaged from the academic role, the expectation of re-engaging with another role to compensate was a struggle, and as such, they reflected on whether they had other skills to engage in a separate role:

> *"What other skills do I have beyond teaching, research, and academia? I've been here for so long, that I don't know what else I could be doing. Am I meant to be doing something else, or do I just relax and do nothing? Take up a hobby? I have no idea." (05).*

Here, the participant questions their skill base and acknowledges tensions with how to navigate life beyond academia. We propose that the women may be feeling vulnerable based on navigating these complexities, by feeling uncertain of what it means to be an individual who has left academia. Broadly, the later-career women academics reflected on what it would mean to identify themselves beyond academia, as well as being an academic, and felt the process was largely unclear and unanswerable based on their positioning at the time of the interviews. Whether they had expressed planning for, and being ready to leave academia, or not, the later-career women academics were positioned in their academic role when we conducted the interviews, and as such, it is unclear whether the women were able to resolve the tensions expressed. It was evident though that the participants were currently engaging in the difficult process of conceptualising their identity beyond academia. One participant summarised the experience of conceptualising their identity, by acknowledging that no matter their career stage, status, or positioning, that it reflects a *"metamorphosis of*

*sorts" (16)*, where identity is forever changing based on their specific context. This was acknowledged to assist in preparing the women for life beyond academia:

> *"We identify differently based on the different times of our careers, with who we interact with, how we view ourselves, and how others view us, within academia. How our outside context intersects can influence how we identify, and considering what is important to us at any given time, I truly feel, can change our identity. . .our identities shift and change, it's a metamorphosis of sorts. That's how we navigate academia. That's how we survive academia." (16).*

## 4. Discussion

Our aim was to explore how later-career women academics conceptualised their academic identities, and our findings reflect four subject positions. First, later-career women academics acknowledged tensions between the academic they wanted to be, versus the academic that the system required them to be. Second, a shift was recognised in terms of how women were evaluated and positioned in relation to their identities, now based on experience and expertise, rather than gender. Third, the later-career women reflected on their academic experience and noted tensions surrounding their decision to continue working in academia, versus considering whether it was time to leave the setting. Finally, the women reflected on what it means to have survived academia, feeling rewarded in their academic achievements and impact, resulting in their disengagement from academia.

Some of the later-career women academics acknowledged difficulties in resisting the Insecure Woman positioning, as well as adopting ways of resisting and critiquing hegemonic practices. Difficulties encompassing resistance have been noted by Foucault, who left the process and how to engage in said resistance up to the individual (Foucault 1980). The difficulty surrounding resistance related to how some of the later-career academics felt uncertain on how to navigate the system beyond ways that were viewed as normalised and viable. Definitions of how academics regulated and reported on their own work (and the work of others) has been defined as an academic norm (Moss-Racusin et al. 2015), where what has been taken to be truth and inevitable in the academic global episteme of power, shaped how the later-career women conducted themselves. How some of the later-career women academics navigated academia appeared constrained by tensions and inconsistencies in meeting the expectations of the academic institutions, as well as the values of the women conflicting with academic discourses. As such, insecurities manifest when the acknowledged, dominant forms of truth and knowledge, and the ways in which later-career academics seek to shape their identities, are inconsistent and contradictory (Parsons and Priola 2013).

Examining the particular roles and identities of later-career women academics assisted us to consider and deconstruct how certain identities were altered by culturally specific and normative expectations. For example, expectations of what a later-career woman academic would do in relation to evaluating one's career, being viewed as the expert in their field, and choices surrounding retirement, were noted in our research, and acknowledged as consistent with what one expects from an individual as they reach the later stages of their lives (Earl et al. 2018). Being able to reflect on such practices and expectations can be viewed as life-giving and enabling, or subjugating and repressive, dependent on the context experienced by the later-career women academics (Thompson 2015). Our participants acknowledged the former (for the most part). These positionings worked to initiate and sustain the level of agency that is experienced, to feel a sense of autonomy over one's decision making, as well as feeling a sense of freedom over how one experienced academia and conceptualised their identities (Burke 2020).

Conversely, discussions with some of the women academics revealed tensions surrounding how the later-career stage posed tensions in both their professional and personal lives, even when discourse of reflection and evaluation were evident. Such tensions appeared to build and manifest according to the changing social, political, and organisational attitudes surrounding older adults, and their positioning within society. For example, older

age has been chronologically considered by individuals and institutions as the period where a decline in work ability and motivations are recognised (Topa and Alcover 2019). While our participants were identified as the expert academics within their field, paradoxically, they were also pressured to exit academia as they were viewed as *'having done all that they could'* within their role. Such assumptions make evident the discriminatory societal norms and attitudes surrounding age within both academia, as well as society more broadly (Topa and Alcover 2019). Higher education institutions need to explore and deconstruct these ageist attitudes surrounding work productivity, and to consider the consequences of work being aligned with one's age, especially at a time where an increase in life expectancy and prevalence of workers aged between 50 and 65 years has been noted (Oplatka 2007).

Implicit within *'making it'* to the later-career stage for women academics was that they had experienced danger and coercion in academia over time, underpinned by patriarchal, gendered, and neoliberal discourses through disciplinary power and surveillance. As a result, many participants mentioned they were constructed and guided to tolerate and survive the system. Embedded within the narratives of our participants were sentiments acknowledging aspects of the academic system which were unsafe, such as the forceful placing of subjectivities onto the women academics, having to meet the expectations of the system with limited resources, sacrificing the sense of self in response to these expectations, and the expectation that women *'have to'* survive and break through the patriarchal, neoliberal, and gendered, heteronormative practices that have disadvantaged them. The acknowledgement of these aspects illuminated the presence of the traditional academic way of being, perpetuated through epistemic violence, where colonisers have created and occupied hegemonic positionings that have enabled them to afford themselves power, which has privileged the knowledge and meanings that are most beneficial for them (Brunner 2021). This was evident through the privileging of white, cis-gender, able-bodied, heterosexual men, and their Eurocentric knowledge, as the normative academic and way of being and thinking within academia. Additionally, the expectation that once the women academics have *'made it'*, that they will exit their role (with the assumption that the quality of their work will decline) is concerning. In combination, these perspectives have worked to evoke structural and cultural violence within the experiences of the later-career women academics, where these mechanisms have been used to justify and legitimise the dominant and privileged academic way of being.

### 4.1. A Nod to Context: COVID-19

It would be remiss for us not to address the COVID-19 pandemic, and its influence on the Australian public higher education setting. The COVID-19 pandemic has transformed our expectations, personal lives, and professional careers in unprecedented ways (Shalaby et al. 2021). As the virus spread, it further exposed the existing structural and gendered inequalities, as well as further deepening others, in societal contexts (McGaughey et al. 2021; Shalaby et al. 2021). Specifically, to our research context, the pandemic has affected both the Australian, and worldwide, university sector (McGaughey et al. 2021). Academics have been significantly impacted, in relation to their capacity to undertake their academic responsibilities, based on the rapid campus closures and shifts to online modalities in response to the pandemic (Marinoni et al. 2021; Watermeyer et al. 2021). For example, engaging in laboratory and field-based research has been noted as a struggle, as well as the disruption of teaching by adapting to online pedagogy in response to social isolation measures and quarantining (McGaughey et al. 2021). Additionally, the management of finances and systems have been particularly vulnerable, whereby the COVID-19 pandemic has further exposed the weaknesses of the dominant neoliberal business model evident in higher education, as well as the placing of social and travel restrictions which have impacted the feasibility of service offerings (particularly to international students; Ross 2020). In response to these changes and impacts, the higher education institutions have initiated voluntary, and involuntary redundancies, with an approximated 21,000 job losses

so far to offset the financial impacts of the pandemic on tertiary education in Australia (McGaughey et al. 2021; Ross 2020).

While these impacts are clearly outlined in the literature base, they are further exacerbated by the operation of gender within higher education, specifically, how women academics are positioned to navigate their personal and professional roles amidst the COVID-19 pandemic (McGaughey et al. 2021). For example, while all academics have suggested tensions surrounding work-related stresses, digital fatigue, concerns over long-term changes to their teaching and research responsibilities, and difficulties managing a work-life balance, women academics have suggested experiencing more of the workload based on the balancing of their work and home lives, as well as the perceived valuing of their roles and contributions being viewed as *'lesser than'* within the pandemic (Gorska et al. 2021; Venkatesh 2020). Additionally, women academics perceived an undermining of academic and personal autonomy during the pandemic, with rigid requirements surrounding their teaching practices, concerns surrounding their competence, and feeling that their professional responsibilities are less valued, compared to men, have been expressed (Crick 2021). Overall, women academics at all career stages acknowledge feelings of abandonment, relatedness, and community amongst academics, at a time where support and collaborations are suggested as needed to assist with navigating the context surrounding the pandemic (McGaughey et al. 2021). These tensions are further exacerbated when considering the home context of the women, with some women academics suggesting that their male spouses view their work role as greater than, or more valued, than the women's working role, even when the women have the same, if not more, responsibilities to balance (Gorska et al. 2021).

While these perspectives are not necessarily *'new'* within the literature base, what is novel, and of importance here, is considering how the existing issues for women within the neoliberal, marketised sector have been exacerbated and compounded (Gorska et al. 2021; McGaughey et al. 2021; Venkatesh 2020). These issues are not gender neutral, with women academics being disproportionately affected by them (Gorska et al. 2021). While there have been clear impacts on the identities and experiences of women academics due to COVID-19, it is important to note that the semi-structured interviews in this study were conducted within the final quarter of 2019, at a time well before the pandemic existed, and impacted on the current context. As such, it is important to consider that our findings are contextually embedded within a pre-COVID-19 setting and should be evaluated as such. This is not to suggest that the experiences and identities of women academics mid- and post-pandemic are not of interest to explore, rather, research conducted thus far has begun to explore how the pandemic has amplified pre-existing issues within the Australian public higher education setting. The exploration of these issues for women academics demonstrates the relevance and applicability of our research conducted pre-pandemic, serving as a qualitative baseline, as well as considering how it is needed within our current context to illustrate the importance of assisting not only women, but all academics, to navigate academia.

*4.2. Recommendations and Reflections*

Practically, the experiences and identities conceptualised and shared by the participants are valuable for the wider community, the research community, as well as policymakers and those who exercise political influence. It is important though, to acknowledge that particular changes and differences in the needs, skills, priorities, challenges, concerns, roles, and relationships of the women academics need to be considered within these implications. As these aspects have been shown both in our findings, and the literature base (Kachchaf et al. 2015; Knights and Clarke 2014; Pick et al. 2017; Pifer and Baker 2016; Rainnie et al. 2013) to evolve over time, differential support for women academics across identities and career stages is necessary and important. Subsequently, our findings reinforce the importance of supporting *'the whole person'*, both professionally and personally, when considering the application of the implications listed next.

First, participants acknowledged the importance of support networks in the navigation of their careers and found it easier to be embedded within academia with these networks being a part of their experience. As such, the importance of networking and mentoring programs was evident. We propose programs that are women-only (to ensure that women can feel safe, interact, and have discussions about their experiences of academia, where they feel women may be the only ones who can understand their experience), and additionally, programs that are inclusive of all genders (to further illustrate and discuss how resolving the inequities is the responsibility of all academics, as well as the system, to consider, rather than just focusing on women's responsibility). We also propose programs that are multi- and inter-disciplinary as well, as the integration of gender, as well as discipline, can be useful as a way of illustrating awareness of the complexities of the academic system. Members could provide examples of how they have progressed, or are progressing through academia, what sort of barriers they might be experiencing, provide support to other academics, disclose the *'tricks or rules'* of the academic game, as well as being able to establish new connections. As such, these new connections, and subsequent knowledge shared, can be a useful tool in assisting all academics in how to navigate an academic career.

With the participants acknowledging issues with the academic system, it is imperative that an examination of the system, and its ways of being and doing, are implemented. As such, we propose that change occurs over time within university policies, guidelines, and protocols. Here, the underlying issues surrounding the organisational culture, context, office politics, and impact of emotional labour would need to be considered. Understandably, changes that are systemic in nature, that problematise the system, are considered easier to acknowledge, but harder to implement (Watzlawick et al. 1974). As such, we acknowledge the following practical implications, which, while ideal, would take time to employ. These suggestions are in terms of creating cultural change within academic institutions and making the setting workable. First, self-care initiatives, such as taking mental health days, or breaks for morning tea/lunch away from the desk during the day, should be introduced into existing policies and guidelines surrounding work practices. Additionally, guidelines surrounding working practices should be implemented, whereby academics can avoid the excessive workload and burden of overworking, and expectations surrounding productivity under less working hours should be more manageable and aligned with the care initiatives that the institution would implement. With participants suggesting a lack of access to opportunities, we propose offering more professional and personal development opportunities, with more support being provided to assist women in developing throughout their careers at different career stages, with a focus on building confidence and efficacy. While some of these opportunities could be individually focused, we propose opportunities for collaboration and networking to be provided also, whereby institutions can actively assist academics to build strong peer support networks via team building activities, and frequent social events to maintain the networks that are created. These opportunities need to be accompanied with the institutional providing of time through reducing workloads for employees to engage in hobbies, exercising, socialising, and other activities that separates the worker from the work, while facilitating connections between employees.

For many of these practical implications to be useful, there needs to be a shift surrounding how we, as academics, view the academic environment and context, as well as the people within it. As such, we propose that the focus is shifted from a deficit to strengths-based perspective. Specifically, rather than considering what might be *'wrong'* with women to not engage in STEMM adequately, we consider how the STEMM institutional environment can remove the systemic barriers that prevent not only women, but all academics with diverse identities and perspectives to engage effectively. This would also involve changing the prototype surrounding what it means to be an academic, by reconstructing the system, and its members, as being more inclusive to the multiplicity of ways to be an academic. Additionally, it would also mean that we would need to develop more of an awareness of the currently existing practices that can negatively impact women and minority groups within STEMM. These practices were suggested to lead to negative mental

and physical health outcomes within our findings, with the climate of STEMM experienced as hostile for the women. Developing this awareness could involve educating all academics on why particular discourse and language is problematic, as well as disrupting the taken for granted assumptions (i.e., the status quo) within academia, to be able to create a safe and collaborative environment for all academics to exist in.

Disrupting these assumptions relies on an awareness of one's individual practices and ways of doing and being as an academic within STEMM. Through legitimising the issue, equality and equity has been suggested as needed within academia, but we question the ways that we acknowledge and suggest how this would be achieved. As such, the implications, and recommendations we suggest are our attempt at proposing what an equitable academic system could look like. We call on all faculty members within STEMM to view our findings as an opportunity to reflect on their own understandings, assumptions, and commitments to promoting gender equity. To engage in this practice, academics may need further opportunities to be able to identify and reflect on their own implicit privileges, assumptions, and biases. Sustained professional and personal development must be engaged in and maintained, as we may be unable to implement transformative change without this consistency. We need to transform the attitudes, privileges, and biases of academics more effectively, by engaging in these practices on a consistent and regular basis.

### 4.2.1. Reformative versus Transformative: Ways to Conceptualise Social Change

Recommendations for changing the aforementioned issues surrounding women's identities and experiences in higher education in the extant literature base have thus far focused on the individual (i.e., women), constructing it as their responsibility to change the system. Where we attempt to mitigate this cycle of individualised blame, is by posing implications and recommendations in relation to how the different orders of change can be utilised, in an attempt to debunk the current way of being which limits change to the social systems that support the relationship between the powerful, and the powerless (Rappaport 1977). In practice, women academics in higher education are often blamed for their victimisation, while the power dynamics that have been identified in our analysis continue to operate unchanged (Ryan 1971). Discourses and practices that underpin individualised blame can be legitimised and viewed as reformative to the system; in response, the interventions can reflect the practice that it attempts to resolve. While reformative strategies may be appealing to individuals at first, as they can reflect immediate gratification surrounding social change, they can retain the traditional hierarchies and practices that currently exist (Mills 2014; Ropers-Huilman et al. 2016). As such, the commentary in this article has acknowledged going beyond observing difference, to be able to intervene and promote equity and transformative, second-order change. It is acknowledged that this is a process which can take time.

In tying the threads together that can underpin recommendations for social change, we acknowledge the role of different orders of change in understanding the issues that surround women's academic identity conceptualisation. The solutions that can be generated from the analysis of individuals, such as women within academia, represent first order change, which can be often based on common sense, which can create and/or exacerbate the issue (Rappaport 1977; Watzlawick et al. 1974). What is important though to recognise, is that thus far, the interventions and strategies that have been designed to support and improve the experiences of women in academia are well intentioned, and important to consider. While important, these efforts need to be extended to create genuine, transformative, second-order change, whereby the system that maintains the traditional way of being is both questioned and challenged, and the individual is looked beyond to consider the collective also (Rappaport 1977; Watzlawick et al. 1974). If the importance of the collective, and the system, are minimised, this can produce decontextualised strategies and interventions that produce a distorted understanding of the social structures that shape unjust relationships, further reinforcing the beliefs embedded in individualised blame, which

neglects the role of the environment (Fine 1986). Within the environment, the standards of the system (i.e., *'rules of the game'*) govern the way that social relationships between women academics and those in positions of power operate. As such, the recommendations that are posed need to be aimed at changing the context, and its rules, rather than simply addressing the individual (Rappaport 1977; Sarason 1974; Watzlawick et al. 1974).

In addressing the role of different forms of social change, we mentioned how interventions suggested thus far were well intentioned. While well intentioned, further to this, we propose that in considering how the state of higher education can be improved, it needs to reflect a gradual process, where transformative change may be the desired change, but we need to first engage in ameliorative or reformative change as a starting point. As such, our recommendations are not intended to solely propose drastic forms of social change, as this may not be feasible within the current context. Rather, we propose recommendations with a comprehensive understanding of the social context, as well as the related structures and relationships that maintain injustice, as without this consideration, it would be a thoughtless endeavour (Fine 1986). Further, in examining the way that academia has operated for women, it is just as important to not only consider how to intervene, but also, where to intervene, and how gradually, or abruptly, the specific change should be introduced (Rappaport 1977). The potential consequences of specific recommendations should also be considered before any action is taken, as failing to consider these consequences may have the unintended impact of further exacerbating the tensions and challenges experienced by women academics.

It is clear then, that the way in which women academics identify within and beyond academia, can have major implications for the recommendations that follow. With such a focus on the individual and blaming the victim, past practices have produced first-order change. For true, reflective, transformative, genuine change to occur, the problems must be reformulated and reconceptualised, where the standards of academia that support interactions between the powerful, and the powerless, are challenged and critiqued. Bishop et al. (2002) acknowledged that while developing self-awareness and an understanding of the social processes and structures in academia (such as victim blaming) is difficult, nevertheless, it is of importance when considering how the proposed recommendations would be implemented. De Botton (2004) further supports this claim, and states that as individuals, we can be so well socialised to particular social forces, that they can often go unnoticed, making them difficult to observe, critique, and challenge as they become so embedded within our society:

> *"The essence of ideological statements is that, unless our political senses are developed, we will fail to spot them. Ideology is released into society like a colourless, odourless gas. It is embedded in newspapers, advertisements, television programmes and text-books—where it makes light of its partial, perhaps illogical, or unjust, take on the world; where it meekly implies that it is simply stating age-old truths with which only a fool or a maniac would disagree."* (pp. 214–15).

Subsequently, it is of importance to have the ability to observe the trends of the obvious (i.e., the status quo) which can be obscured by dominant worldviews, as well as having a critical understanding of the social context (Bishop et al. 2002).

### 4.2.2. Tying It All Together: Recommendations for Social Change

This research has a crucial role in being able to raise awareness amongst researchers, scholars, disciplines, professions, system facilitators and users, and broader society in determining the complex social processes that impinge on the experiences and identities of women in academia, as identified in our analysis. Looking beyond the individual and ensuring we attend to all aspects of the social context can raise new questions. Being able to accept the powerful role of the context (in relation to values, worldviews, and norms) in how academic identities are conceptualised for women in academia, can encourage us to consider what an equitable allocation of power would be, allowing the opening of new possibilities (Fine 1986). We would recommend for individuals to be critical of

the current systems of possibility, rather than just passively accepting the status quo as a given (Fine and Asch 1988). Such forms of critical questioning and reflection should be encouraged within the context of reflexive practice, mentoring, and educational programs. Further incorporating the voices of all academics in the future would be beneficial for the professional discourses and understandings surrounding identity and equity in academia to be challenged.

When considering the use of discourse surrounding a work–life balance, re-conceptualising this *'balance'* into an integration of work-in-life, or work being one component of one's life out of many, is important. Experiences in life are not separate, and as such, cannot be implied by balance, as neither family, caring responsibilities, hobbies, or careers are static (Clegg 2008). Further, reconsidering the *'pipeline'* and constructing it as a life-course instead would be useful. The *'pipeline'* metaphor reflects masculinised orientations of a linear career advancement and progression over time, where there is one entry point and no ability to be able to exit and re-enter (Shaw and Stanton 2012). This does not encapsulate the many diverse experiences and progression of women academics, and as such, life-course would be more encompassing of the diverse experiences of all academics. Life-course refocuses the work-life commentary to consider that branches are available for entry, and re-entry. Our findings illustrate the benefit of considering these experiences and identities under a life-course perspective, as the way that the women academics were influenced by their changing social and historical contexts impacted the decisions they made, based on the opportunities they were provided, and the constraints of the social structure and culture of higher education. Finally, moving beyond constructions of women academics' experiences and identities being focused on family, to being more focused on life, would be more inclusive of the many diverse constructions of life that recognise women's multiple ways of being.

Comparing the narratives across each of the career stages reflects the necessity of a broader life-friendly perspective (rather than just family friendly) when considering the conceptualisation of policy. As individuals are living and working longer, as well as the shift demographically in our aging country (Allen et al. 2021; Ropers-Huilman et al. 2016), higher education institutions need to consider embedding their policies in the view of human lives as progressive and ongoing, to include the complexities and nuances surrounding intergenerational families, caring responsibilities, and other outside of academia interests and responsibilities. As such, an inclusive approach is important, not just considering the academic women in our research, but for all academics Australia, and worldwide, to benefit all and allow for a more holistic perspective towards careers, life, and beyond. The importance of addressing this changing perspective surrounding work and life is evident, especially considering how academics nowadays are seeking a meaningful life both inside, and outside, of work (Philipsen et al. 2017).

All academics (not just women) need to collectively consider and debate the direction that their higher education institution is taking, whether the direction is in the best interests of both women, and society, and whether there are any solutions that are being posed that would be effective in considering genuine, transformational change. While conceptualising the debate, and how this could take place, is beyond the scope of this article, the discussion would likely be complex as it would attempt to challenge the individualistic and neoliberal norms for which academia is known, as well as the specific discourses that shape the status quo. We would expect that the implications of women feeling fearful, uncertain, and unsure of making themselves and their stories heard, for themselves and society, would be considered, and debated, not just by the women themselves, but by the broader higher education sector. Responsibility lies with the academy as the employer to both participate in, and contribute to, the resolution of the aforementioned issues as the narratives of the women suggest that the current workplace practices and policies are contributing to problems relating to the long-term health and well-being of women, as well as the overall sustainability of the higher education setting.

Additionally, with the intent of collaborations and ongoing debate, discussions should be engaged in to allow for the collective resisting and challenging of the systems and practices that contribute to women's disempowerment in academia. This could include, for example, the establishing of collaborative research networks, collective research submissions, and spaces provided by employers (to include temporal, physical, and financial resources) to meet, discuss, and debate about the issues within academia. Ways for collective debate between the hierarchies in academia, and academics, should be found that allow for the stories of all academics to be heard. Some of the women in our research acknowledged feeling fearful of negative repercussions if they decided to speak out against other academics, and the system, and/or feeling anxious if they considered ways of navigating academia that may compromise meeting the expectations of their employment. Therefore, such avenues need to be safe and secure in terms of employment security, as it has been considered by our participants as risky and dangerous for women academics to speak out.

### 4.3. Implications and Strengths

Theoretically, our research adds to the understanding of what it means to be a later-career female academic, and how the construction and navigation of identity is an ongoing and fluid process, influenced by one's personal and professional experiences, which demonstrates the interplay of dynamics that are specific to the later-career women's academic context. Extending on this dynamic, our research is the first to suggest specific subjectivities and subject positions for later-career women academics, in terms of how they can be positioned to occupy and engage in specific roles and responsibilities. Furthermore, in drawing on the Foucauldian approach in our research, this has illuminated specific subjectivities and positionings made available to the later-career women academics, as well as recognizing the political, historical, and social influences that have emerged over time to impact on these women. In dedicating ourselves to the critical, Foucauldian perspective, we have exemplified the explicit relationship between knowledge, power, discourse, subjectivities, identities, and how particular roles are legitimised in Australian public higher education.

### 4.4. Limitations and Future Research Directions

Our findings are only representative for later-career women academics in the Australian public academic context. As such, caution needs to be applied when considering our findings to other contexts (Lincoln and Guba 1985). Future research could explore the conceptualisation and construction of academic identities within other socio-cultural contexts, academic faculties, private academic settings, and geographical and cultural contexts, to consider the diverse needs, experiences, ways of being, and individual perspectives from different settings. Our research project did exclude other academics who fell outside of the inclusion criteria, such as gender-diverse academics, and male academics. We propose that future research be conducted to explore (1) the way male academics at different career stages construct their academic identities, (2) how male academics perceive gender to be embedded within higher education, and (3) how male academics perceive their positionings and ability to navigate the academic setting, compared to other academics. Finally, it is important to recognise that the renegotiation of subjectivities and identities for academics is important, for all to thrive within the system; therefore, collecting the perspectives of all academics within research, but additionally, the academic institutions also, at each career stage, is of importance. We extend on our early, and middle-career women academics studies (Phillips et al. 2023a, 2023b), to propose that future research explore the academic experiences and identities of women who have since left and/or retired from academia.

### 5. Conclusions

The becoming of, and being, a later-career woman academic is marked by an increase in administrative and leadership responsibilities, as well as being positioned to play a key role in the operation of the academic institution. Tensions emerge when the later-career women academics are expected to balance these expectations, while simultaneously

contemplating how they choose to remain, work, and identify within academia. Our findings illustrated that while the later-career women academics experienced a shift in how they were evaluated in academia (based on expertise, rather than gender), tensions arose when the women began to reflect on what it means to have survived the academic setting. While some experienced insecurity, others were positioned as experts; either way, the women began to reflect on what, and how, they have contributed to the academic setting. Subsequently, the findings reflected a questioning of what it means to have been a part of academia, what it means to identify as a later-career woman academic, as well as evaluating what it means to identify beyond the academic setting.

**Author Contributions:** Conceptualization, M.J.P. and P.L.D.; methodology, M.J.P. and P.L.D.; formal analysis, M.J.P.; investigation, M.J.P. and P.L.D.; data curation, M.J.P.; writing—original draft preparation, M.J.P.; writing—review and editing, M.J.P. and P.L.D.; visualization, M.J.P. and P.L.D.; supervision, M.J.P. and P.L.D.; project administration, M.J.P. and P.L.D. All authors have read and agreed to the published version of the manuscript.

**Funding:** This research received no external funding.

**Institutional Review Board Statement:** The study was approved by and conducted in accordance the Human Research Ethics Committee of CURTIN UNIVERSITY (protocol code HRE2018-0606 and HRE2018-0606-01, 13/09/18) for studies involving humans.

**Informed Consent Statement:** Informed consent was obtained from all subjects involved in the study.

**Data Availability Statement:** The authors report that the data is stored on a password protected institutional research drive, only accessed by the research team.

**Acknowledgments:** We acknowledge the support and mentoring of Lynne Roberts during this research project and thank her for her assistance.

**Conflicts of Interest:** The authors declare no conflict of interest.

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
