# Peer review of "“I’m an Academic, Now What?”: Exploring Later-Career Women’s Academic Identities in Australian Higher Education Using Foucauldian Discourse Analysis"

_socsci, doi:10.3390/socsci12080442_

Round 1
Reviewer 1 Report
It's a real issue that affects women in academia, but can also affect other categories in other work environments.
What recommendations would you make following these interviews and results?
How can we ensure equality between men and women in the various sectors of activity?
Could you suggest future research in developed and developing countries?
Thank you for addressing this type of issue, which concerns all working women.
Reviewer 2 Report
the research method is appropriate and the writing is good.
Minor writing items:
line 679-680 "so I lowkey don't really care" - is 'lowkey a correct word?
other than the one note about the use of 'lowkey,' the quality of English language use is good
Reviewer 3 Report
Overall, the paper is interesting to read. Some elaborations are needed to make it regarding some issues.
Table(1) contains many important information. The authors need to elaborate further on the information in this table. Some of the information could be the (reason) of differences when answering the questions of the interviewers. For example, the significance of (Current Institution), (Current Faculty), (Current Position).
In 3.4 Materials, It is not enough to just say (An example question was). I think, if possible, all questions should be listed here, or most of them.
Explain more about what is meant by (Semi-structured interview). Provide some limitations that might exist with this approach. Since (later, feedback was provided by email), some elaborations are needed. Since the approach changed. Why social media way was not utilized?
In the Results section, again, not much is said regarding the information in Table 1. How the information affected the results. In my view, this is a major challenge in this paper.
Reviewer 4 Report
The authors of "“I’m An Academic, Now What?”: Exploring Later-Career Women’s Academic Identities in Australian Higher Education Using Foucauldian Discourse Analysis" have done quality research on interlocking topics (later-career academics most generally) that are interesting and important in my view. This paper can contribute to the literature in several ways and in multiple areas. The literature review is sound and coherent, the design is appropriate and the methods are applied well, and the conclusions seem reasonable based on the results presented. I have a few suggestions for improvements, though I believe the manuscript is rather solid at this point:
1. Clarify in the literature if there has been much work with male late career academics, as I am not really aware of any. Thus, this seems to be a broader gap than is being presented - so perhaps a bit more literature review/discussion is needed.
2. Why is 15 years the cut-off for "late" career? I have often considered there to be early, mid and late career phases, and if the average career is roughly 30 years (for those who last beyond a few years), then perhaps 20 years post PhD is most appropriate? Some deeper justification would help here.
3. How has later career academic subject-position been impacted by dramatic shifts in higher education in recent years related to corporatization of the university, modality shifts, COVID/other external threats, etc.?
4. I believe some more specific discussion of how the analyses were conducted would be beneficial, given there are so many different analytic approaches being undertaken in qualitative research today.
5. What about overlaps between the four subject-positions? Is there space to discuss those as well?
